# Holocene Depositional History of Low-Lying Reef-Rim Carbonate Islets of Fakarava Atoll, Northwest Tuamotu, Central South Pacific

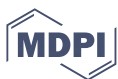

Lucien F. Montaggioni [1,*], Bernard Salvat [2], Edwige Pons-Branchu [3], Bertrand Martin-Garin [1], Arnaud Dapoigny [3], Éric Brunaud [4], Gilbert Poli [5] and Miri Tatarata [6]

[1]  Aix Marseille Univ, CNRS, IRD, INRAE, CEREGE, 13331 Marseille, France; bertrand.martin-garin@univ-amu.fr

[2]  PSL-École Pratique des Hautes Études, UAR 3278, EPHE, CNRS, UPVD, Criobe, Labex Corail, Université de Perpignan, 66860 Perpignan, France; bsalvat@univ-perp.fr

[3]  LSCE–IPSL, CEA, CNRS, UVSQ, Université Paris-Saclay, 91198 Gif-sur-Yvette, France; edwige.pons-branchu@lsce.ipsl.fr (E.P.-B.); arnaud.dapoigny@lsce.iplsl (A.D.)

[4]  Independent Researcher, 44470 Carquefou, France; eric.brunaud@univ-nantes.fr

[5]  Independent Researcher, 75016 Paris, France; poligilbert2@gmail.com

[6]  Délégation à l'Environnement, 98713 Papeete, Tahiti, French Polynesia; miri.tatarata@gmail.com

*   Correspondence: montaggioni@cerege.fr

**Abstract:** Assumptions about the fate of low-lying coral reef islands (atolls) facing global warming are poorly constrained, due to insufficient information on their depositional history. Based on the U/Th dating of 48 coral clasts, the chronostratigraphic analysis of excavated sections through rim islets (*motu*) at the windward and leeward sides of Fakarava Atoll (Tuamotu, French Polynesia) reveal that the deposition of coral detritus started approximately 2000 years ago. Most of these deposits lie on conglomerate pavements or reef flat surfaces, and are about 4500 to 3000 years old. The islet expansion at the windward sites seems to have operated coevally across the reef rim, from the ocean-facing shore lagoonwards. Meanwhile, well-developed, continuous, elongated, vegetated islets mostly occur along the windward, northeast to southeast coast, and isolated islets, vegetated or not, associated with the dense networks of conglomerates, are common on the leeward, partly submerged, western rim. Islet accretion on the windward rim sides is believed to have been mainly triggered by winter storms and occasional cyclonic events, whilst the leeward atoll parts were most likely shaped by distant-source swells from mid to high latitudes. The projections of the accelerated sea level rise in the future suggest that the long-term islet stability at Fakarava could be altered because the islets have accreted under the conditions of the falling sea level.

**Keywords:** atoll; islet deposition; marine hazards; Tuamotu; French Polynesia; Holocene

## 1. Introduction

Low-lying coral reef islets are regarded as among the most vulnerable coastal geomorphic features to climate change [1–7] due to their low elevation above the present mean sea level (pmsl), usually peaking at maximum elevations of 3 to 5 m. A future rise in the sea level and increasing marine hazard intensity are postulated to exacerbate the vulnerability of coral reef islets, posing critical societal challenges [6,8–12]. In the South Pacific, marine hazard events, e.g., storm surges, cyclonic and depressional waves, lows, distant-source swells, and large astronomical tides, can generate large local wave heights, resulting in coastal flooding [13–18]. Such marine hazard events have a significant impact on reef islets, including both construction and erosion [12,19].

However, the future of tropical atoll islets facing the anthropogenically driven climate disruption remains uncertain globally. A variety of scenarios have been proposed:

islet persistence, with possible lateral migration [20–33], islet accretion [30,34–36], or islet erosion [3,6,37–45].

This uncertainty is mainly due to the limited range of studies isolating the main controls of coral reef islet building and plasticity [19,23,25,27,46]. Detailed knowledge of the multi-millennial development history and variability of coral reef islands is fundamental to be able to separate the sea level rise from other marine hazard forcings and to evaluate the future physical behavioral responses of islets.

Atoll islets in the Indo-Pacific, known as *motu* in French Polynesia, have been interpreted as landforms deposited since the Mid Holocene, i.e., the last 6000 years, in relation to sea level change [19,36,37,46–51]. The interplay between the course of sea level, reef growth and clast supply from adjacent reef sedimentary sources was assumed to be under the critical control of islet accretion and sustainability [37,49,52,53]. Islet initiation and expansion have occurred during a rise in the sea level or stillstands higher than the one at present [12,36,48,49,52,54–58] or during a relative sea level fall [19,36,37,48,49,52,57,59,60]. These studies strongly suggest that reef islets are able to form at different sea level positions. There are clearly both spatial and temporal variations in the conditions of islet settlement and accretion from region to region and within the same region. Storms and cyclones are also known to play a significant role in atoll islet shaping [9,46,61–68]. In the Tuamotu, previous detailed reconstructions of *motu* development across millennial timescales revealed that marine hazard events were the main drivers, not sea level changes [12,58,60,69]. By contrast, in the Maldives, the Western Indian Ocean sea level was demonstrated to be the most critical control of atoll islet formation ([52] and references herein). Such a discrepancy could be explained by the differences in regional climate regimes: the Tuamotu Archipelago belongs to an area suffering occasionally cyclones, tropical depressions, and remotely generated swells, with 3 to up to 15 m high waves [13,14], while the Maldives are a cyclone-free region, but can be occasionally impacted by windstorm waves about 5 m high [1] and swells propagated from the south, not exceeding 1.9 m high [53].

The objectives of the present study are to describe the internal structure of some rim islets and to reconstruct their development history at Fakarava Atoll, northwest Tuamotu, French Polynesia (Figure 1a). The chronostratigraphy of windward, northeastern, and leeward, southwest sedimentary bodies was established from excavated sections, based on the uranium series dating of 48 coral clasts. This resulted in the identification of a new accretional model of atoll rim islets, evaluated within the framework of the change in the sea level and of the frequency of marine hazard events. This model is compared with other previously described islet growth scenarios in the Tuamotu. The present study sheds light on the significant differences in the timing of islet initiation and in the accretional modes from atoll to atoll. The examination of the zonal differences in islet initiation and development within the considered region is of prime importance to better recognize the drivers of islet evolution at a local level. This is critical to understand how these *motu* will naturally adapt to climate-related disruptions and thus allow for human settlement to be maintained. Indeed, Fakarava has about a thousand inhabitants living in the village of Rotoava in the northeast of the atoll. Tourism is an essential local economic resource, especially including lagoon and intra-passe diving. Note that this atoll is one of the UNESCO Biosphere Reserves in French Polynesia, together with some nearby atolls.

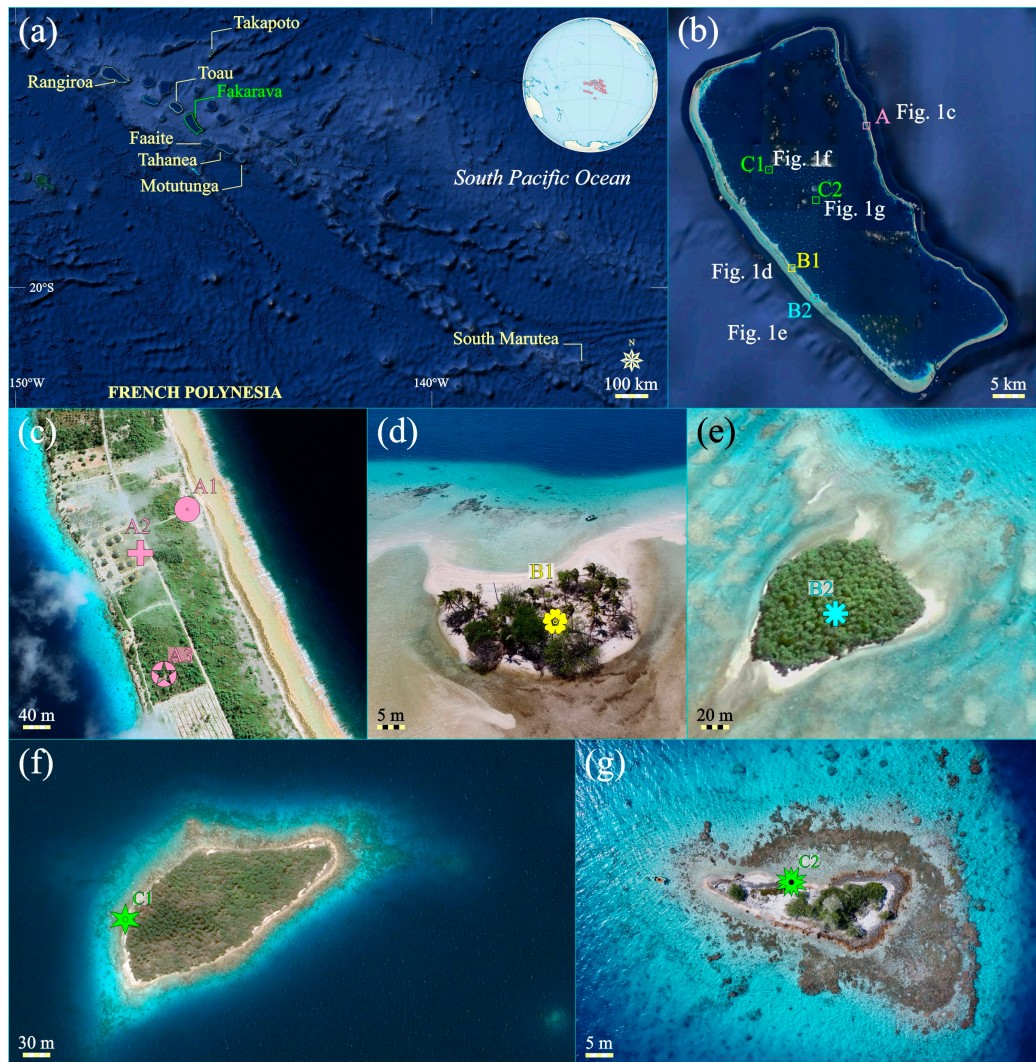

**Figure 1.** (**a**) Location map of Fakarava Atoll in the Tuamotu Archipelago, French Polynesia, central South Pacific; (**b**) Google Earth map of Fakarava, with location of the selected site areas: A: windward, northeast islet (**c**) with location of the three stratigraphic sections A1, A2, and A3; B: leeward, southwest islets B1 (Motu Topikite, (**d**)) and B2 (Motu Otekofai, (**e**)) with location of the studied sections and sampled areas; C: lagoonal reef patches C1 (**f**) and C2 (**g**) with location of dated coral-clast samples (open circle symbols).

## 2. Environmental Setting

### 2.1. Atoll Location and Morphological Attributes

Fakarava Atoll is sited in the northwestern part of the Tuamotu Archipelago, French Polynesia, in the central South Pacific, at 16°18′ S and 145°36′ W (Figure 1a). Comprising 77 low-lying islands (e.g., atolls), this archipelago extends over about 1500 km through northwest to southeast-oriented linear volcanic chains [70]. About 55 km long and 25 km wide, rectangular in shape, Fakarava is elongated along a northwest to southeast axis (Figure 1b), consistent with the general orientation of the volcanic strings. As the second largest atoll in the Tuamotu, it covers a total marine area of 1246 km². The atoll rim is typically asymmetrical: it is widest on its western side (1350 m) and narrowest on its northern (750 m), southern (about 450 m), and eastern (250 m) sides. The total emerged land area, composed of low-lying, densely vegetated, mixed pebble–sand islets is about 27 km² [15], mainly located along the north to eastern reef-rim zones as continuous, elongated deposits, reaching about 2.50 to 4–5 m in maximum elevation. These are locally punctuated by ocean–lagoon exchange channels—*hoa* in the native language —specifically,

at the south–southeastern reef-rim end. The northwestern part of the reef-rim exhibits a series of some 25 islets with surface areas ranging between a few thousands square meters to 10,000 m². Further south, islets are much more scattered and usually smaller. The reef-rim is locally interrupted by passes (Garue and Temakohua) respectively located at its northern and southern ends. Oceanwards, these reef-rim zones are fringed by reef flats not exceeding 40–50 m wide.

By contrast, the southern and western reef-rim zones exhibit open, tidally exposed reef flats, covered by several tens of isolated *motu*, i.e., small, elongated, dome- to crescent-shaped sandy cays (non-vegetated) or islets (vegetated), mainly located in the innermost parts of the reef flats. The outer and mid reef flat zones are occupied by dense networks of unconsolidated to lithified rubble sheets, oriented parallel or perpendicular to the reef front line. The lithified rubble sheets form usually continuous to partly eroded conglomerate platforms, up to 0.50 m thick. There are two distinct bands of unconsolidated rubble sheets and conglomerate pavements, respectively located at a distance of 300–350 m and 800–900 m behind the reef front line. These bands are approximately 30 to up to 250 m in width. Cays and islets occur just behind the innermost rubble band as 30 to 150 m wide units. The outermost part of the western reef flat zones consists of an about 40 m wide algal crest. Inwards, there is a zone of scattered coral heads in front of cays and rubble sheets.

At the regional scale, the lagoon is one of the deepest, with a maximum depth of approximately 60 m, and one of the greatest, with a surface area of 1121 km². There is a high concentration of buildups in the form of pinnacles (height > width) or reef patches (height < width), especially in the central western parts—about 10–12 per km². The total number of buildups can be estimated at approximately 800. Sub-circular to elongated in shaped, their size ranges from 15 m to about 200 m—mean size: 50 m. These buildups rise from depths of 20 to 60 m. Their submarine flanks are usually very steep (>50°) and colonized by dense coral communities that are dominated by poritids and acroporids or cluttered with coral detritus. Three types of buildups are identified according to their top position relative to sea level: (1) submerged, at depths of several meters below present mean sea level (pmsl); (2) subtidally positioned, just below low water spring tide; (3) emerged, at elevations of a few decimeters to up to 1 m above high water spring tide. These buildups at top exhibit different types of geomorphic features, varying in size. The smaller can be occupied by exposed, fossilized, more or less scattered coral heads, related to a mid-to-late Holocene sea stand higher than present [71] or by sandy cays totally cleared of vegetation or with low-growing plants, often transitory, sometimes shrubby. The larger are commonly overtopped by basal, coral conglomerate platforms, up to 1 m thick, usually colonized by dense, shrubby to high-growing vegetation, including remains of high-value, indigenous plants. These are referred to as *wooded islets*. Relatively numerous on Fakarava, wooded islets are apparently typical of lagoons from the Tuamotu atolls as compared to other Pacific atolls.

### 2.2. Climate and Marine Hazard Events

The climate regime in the northwestern Tuamotu region is driven by the combined effects of the trade winds and El Niño-Southern Oscillation (ENSO) which regulate tropical to extra-tropical storms [72]. Like the other northwestern atolls, Fakarava is subjected to trade winds blowing from the eastern sectors for about 70% of the year and from the SE for about 20% (Figure 2). The warm, rainy, austral summer lasts from November to April and the relatively cooler and dryer austral winter, from May to October. During the summertime, the trade winds are weak and generate low wave heights. However, this regime can be altered by waves from tropical cyclones or from distant-source storms forming in the northern latitudes. Fakarava is relatively protected from northern distant-source waves by northern atolls due to the atoll 'shadow effect' [72]. Cyclones are active usually during the hot season, with a large annual variability mainly related to ENSO. The eastward migration of warmer sea surface temperatures during El Niño phases promotes the displacement of tropical storms further east into the Tuamotu region. The highest activity time relates to

February with 40% of events. The cyclone track pattern in French Polynesia is typified by a tapering channel between the Society and the Austral Archipelagos, through which 70% of the cyclone tracks pass [73].

During the austral winter, the region is crossed by strong distant-source wells originating from the south and southwestern latitudes (>40°) (Figure 3a). The related wave regime is typified by periods greater than 12 s and heights of up to 4–5 m [14,74]. Episodes of low wave heights (around 1.5 m) and calm occur between intervals of moderate to high swells (2.5 m). Higher amplitude waves (up to 3.75 m) can be generated by easterly winds and distant-source cyclones [72]. During the summer, swells are similar in direction, period and height to those generated by the trade winds (Figure 3b).

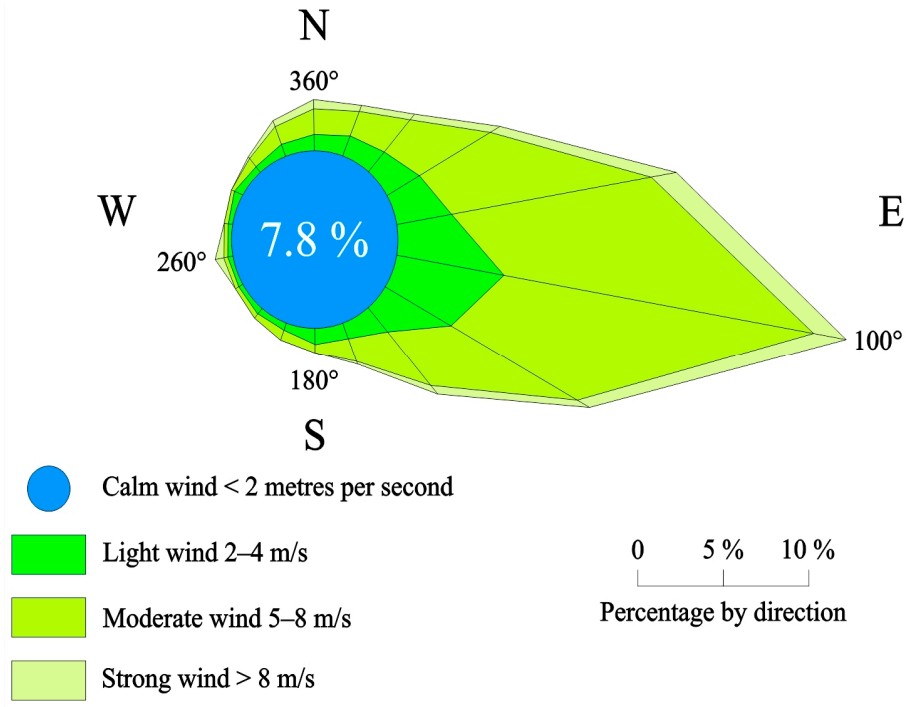

**Figure 2.** Plot of average, annual directions, and speeds of winds recorded as blowing from the northwest Tuamotu region—meteorological station of Takaroa Atoll—from 1981 to 2017. Adapted from [75].

At Fakarava, the air temperature varies between 22 °C and 31 °C. The rainfall approximates 1500 mm/year, with values ranging from 900 to 2200 mm. The tide regime is microtidal and semi-diurnal, averaging 0.5 m in amplitude to a maximum of 0.65–0.70 m at spring tide.

Extreme marine wave hazard events have been poorly documented in the Tuamotu region during historical times; the observational period spans approximately the last 200 years. Based on the analysis of reef sediment archives, cyclogenesis seems to have been of a greater magnitude during the past than during the last century [63], especially during the Little Ice Age in the tropical Pacific [76]. Only about 24 cyclones have been recorded in the Tuamotu for the past 192 years, between 1822 and 2014. Since the beginning of the 20th century, the northwestern Tuamotu region has been hit by six cyclones and a strong tropical depression. As with other Tuamotu islands, Fakarava is rarely impacted by tropical cyclones and depressions [77] (Figure 4). The last tropical cyclone that affected Fakarava was Orama (category 3: wind speed from 178 to 210 km/h; wave heights from 2.7 to 3.7 m) in February 1983. Over periods of several centuries, the number of cyclonic events in northwest Tuamotu have probably not exceeded 2–3 per century (Canavesio, 2014). Gaps of longer than 100 years are not uncommon between extreme wave events. Cyclone hazard records reveal a 50-year return period for waves exceeding 12 m high [13].

Distant-source storm swells are also active in Tuamotu, generated from the northern or southern Pacific high latitudes [14]. As it is subtidally positioned, the reef flat zone from the western rim sides at Fakarava can be easily overwashed by strong waves, thus facilitating great water influx in the lagoon. The sea level could probably rise up to 2.50 m in the lagoon, as observed on Makemo Atoll [14]. Fakarava is also known to experience the effects of low to moderate climate events, such as tropical lows, able to generate strong swells. At Fakarava, such events have resulted in shoreline erosion and retreat [15].

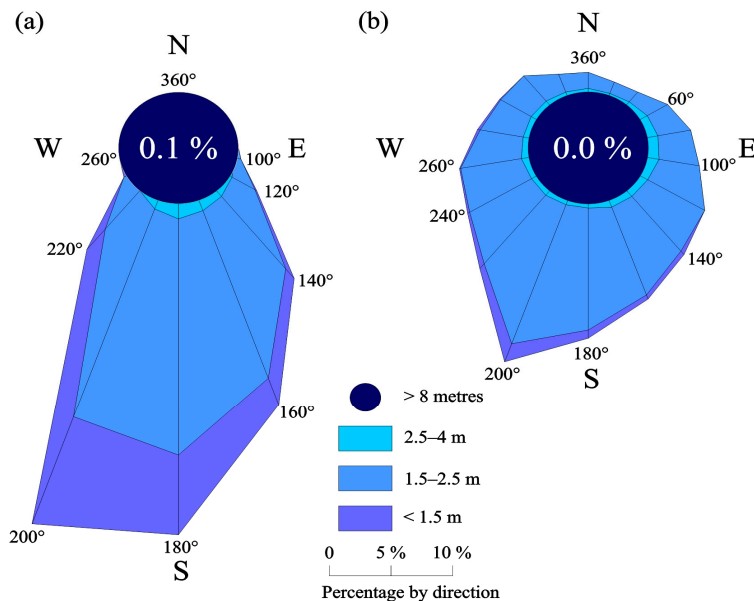

**Figure 3.** Plots of average directions and speeds of swells as recorded coming from the northwest Tuamotu region—meteorological station of Takaroa Atoll—from 1981 to 2010. (**a**) Winter season (June, July, August); (**b**) summer season (December, January, February). Adapted from [75].

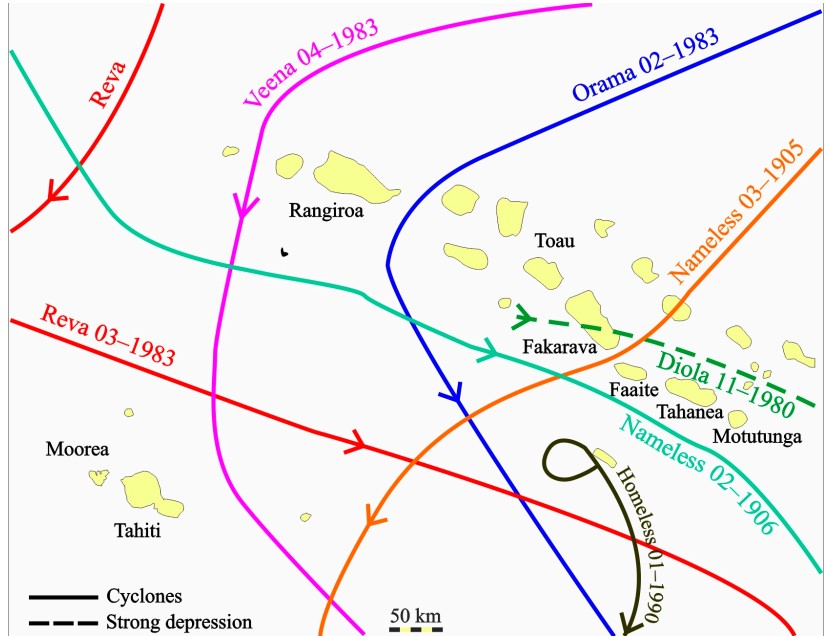

**Figure 4.** Trajectories of tropical cyclones and depressions within the central French Polynesian region since the beginning of the twentieth century. The cyclone trajectories are systematically oriented southwards. Adapted from [75].

Little is known about the role of cyclogenesis and storminess in shaping atoll morphology. In 1906, on Anaa Atoll, cyclone-generated waves removed the western side of the island over a distance of more than 300 m [13]. During Cyclone Orama, in February 1983, the village at Takapoto Atoll, located approximately 200 km north of Fakarava, was also affected, and flooded by 4–5 m high waves. [78] pointed out that about 50–100% of the living fore reef coral communities were destroyed along the eastern side of the atoll at that time. Similarly, little is known historically about the frequency and effects of tsunamis on Tuamotu [79]. As emphasized by Goff et al. [80], Pacific Islands have usually short historic records (<150 years) of tsunami events. The available data suggest that, since the 16th century, the Tuamotu islands were affected by less than ten tsunamis generating 0.3 m to 1.9 m high waves [63]. This was recently confirmed by numerical modelling of the spatial extent and wave amplitude generated by earthquake-related tsunamis in the Central Pacific [81]. Triggered by earthquakes from the Chile, Peru, and Tonga plate boundaries, the propagated waves across French Polynesia range between <1 m to about 2 m in maximum amplitude. The Tuamotu region seems to have not been significantly affected by tsunami waves compared to the high volcanic French Polynesian islands [82].

Absolute sea level changes were reconstructed in the tropical Pacific over a 60-year period (1950 to 2009) by Becker et al. [83]. The sea level was estimated to have risen by about 2.5 mm/yr, a value higher than the mean global absolute sea level rise (1.2–1.8 mm/yr) inferred for the 20th century [84,85]. For the next few decades, models have predicted an acceleration of the sea level rise in the northwestern Tuamotu atolls, with rates greater than 8 mm/yr [86].

## 3. Material and Methods

### 3.1. Field Survey

The present contribution is the result of field trips conducted during the years 2006, 2008, 2019, 2021, and 2022 on Fakarava Atoll. Topographic surveys were carried out at different selected sittings, e.g., islets and lagoonal reef patches, along transects (Figure 1b). Ground levelling was conducted using a conventional automatic level. Along the transects, each reference point was positioned using DGPS coordinates. Elevations were established by reference to present mean sea level, assuming a conservative error of ±0.10 m. In addition, internal stratigraphy and lithology from atoll rim islets was established from analysis of vertical sections in excavations dug by backhoe or in open quarry. Field grain size analysis of coarser detrital material was performed using the Udden–Wentworth classification [87] as measured along the longest axis from photographed square-meter quadrats. In addition, at all selected sites, coral clast samples were collected from walls of cross-sections and adjacent or underlying antecedent substrates for radiometric dating. These allow for 2D chronostratigraphic profiles of the *motu* interiors to be reconstructed. Vertical and oblique aerial images were captured using a mini-drone type DJI equipped with two stabilized, 12 million-pixel-resolution cameras.

The selected windward site (see Area A, Figure 1b) is located in the east–northeast portion of the about 50 km long, continuous *motu* occupying the total area of the eastern side of the atoll reef rim (Figure 1c). Behind a 55 m wide reef flat zone, the emerged *motu* stretches over 245 m from the ocean-facing shoreline to the lagoon margin, from 16°10′48″ S–145°34′05″ W to 16°10′52″ S–145°34′13″ W. Islet topography is typified by a 20 m wide rubble ridge with steep (up to 40°) faces seawards, about 3 m high relative to pmsl. This overlies a conglomerate outcrop, approximately 20–25 m in width. Behind the ridge, there is a gentle-dipping bench. The bench elevation decreases regularly to reach 1.15 m at the lagoonal margin. The bench ends into a less than 20° sloping beach, about 4–5 m wide, resting on an about 10 m wide conglomerate platform. The *motu* is densely vegetated. Sediment sequence analysis and sample collection were conducted at three subsites: from the face of the open quarry (Site A1) at the ocean-facing islet side and from two vertical

excavations dug using backhoe, respectively, through the middle (Site A2) and inner (Site A3) parts of the bench (Figure 1c).

The selected leeward sites (B1 and B2) are located at the mid part of the western, partly submerged reef-rim (Figure 1b,d,e), dotted with about a dozen moving, sandy, non-vegetated to relatively stable, vegetated islets at its innermost border. This zone also displays large, discontinuous rubble sheets, locally consolidated as conglomerate pavements (Figure 5). Two islets were selected to reconstruct their subsurface lithology and stratigraphy, respectively referred to as Motu Topikite (Site B1), at 16°21′53.14″ S–145°41′09.52″ W and Motu Otekofai (Site B2) at 16°25′22.67″ S–145°38′13.71″ W. Motu Topikite is crescent-shaped, about 50 m long by 25 m large, about 2000 m² in surface area, sparsely vegetated and culminating at approximately 1.20 m above pmsl. It is bordered by a 90 m long, 55 m wide conglomerate platform on its ocean-facing side and rests on a reef flat surface in its back face. Motu Otekofai is about 150 m long and 110 m wide, 15,070 m² in surface area. It reaches 1 m in maximum elevation. Densely vegetated, it is covered by forest species—*Cocos nucifera*, *Calophyllum inophyllum*, *Pandanus* sp., *Guettarda speciosa*—together with low-growing *Heliotropium foertherianum*. One excavation was dug by hand using pick and shovel through the center of each islet, down to sea level (Figure 1b,d,e).

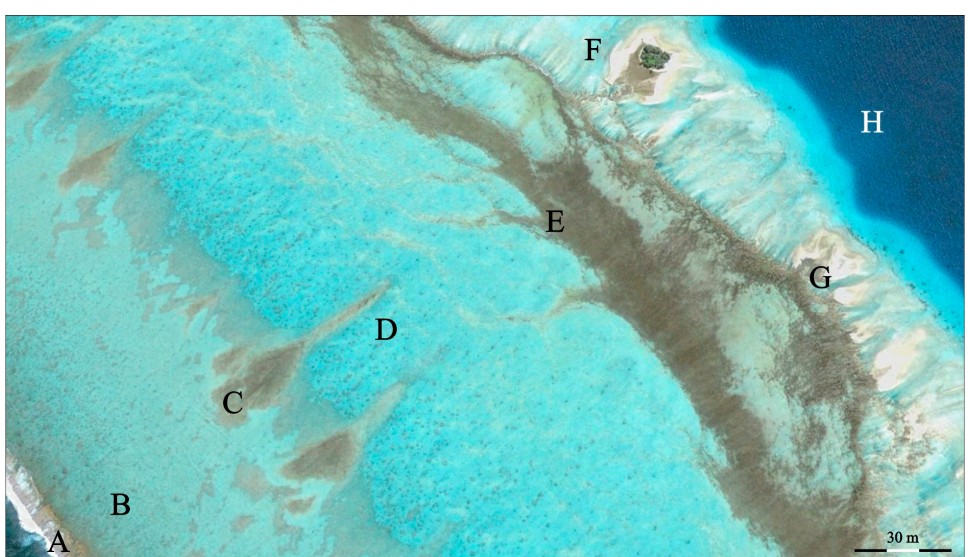

**Figure 5.** Google Earth partial view of the leeward, southwestern Fakarava reef-rim from the outer front line to the lagoon, showing different geomorphic features; A: outer algal crest; B: outer reef flat zone; C: bands of rubble sheet and conglomerates morphologically organized in response to the dominant swell regime; D: inner reef flat zone; E: networks of rubble sheet and conglomerates; F: vegetated islet; G: sand sheets and emerged cays; H: lagoon.

Two reef patches, the emerged tops of which are covered by vegetated (*wooded*) islets and identified as Reef Patch A (Site C1) and Reef Patch E (Site C2), were selected from the lagoon (Figure 1b,f,g). Densely vegetated, Reef Patch A (16°14′46.25″ S–145°42′30.67″ W), known as Motu Hakono, is about 30,000 m² in surface area (Figure 6). Sparsely vegetated, Reef Patch E (16°17′11.69″ S–145°38′22.65″ W) does not exceed 3000 m². Both are overtopped by discontinuous conglomerate platforms, averaging 0.40 m in thickness, and locally reaching about 1.10 m in maximum thickness. Their maximum elevation ranges between 1.0 and 1.30 m (Figure 7).

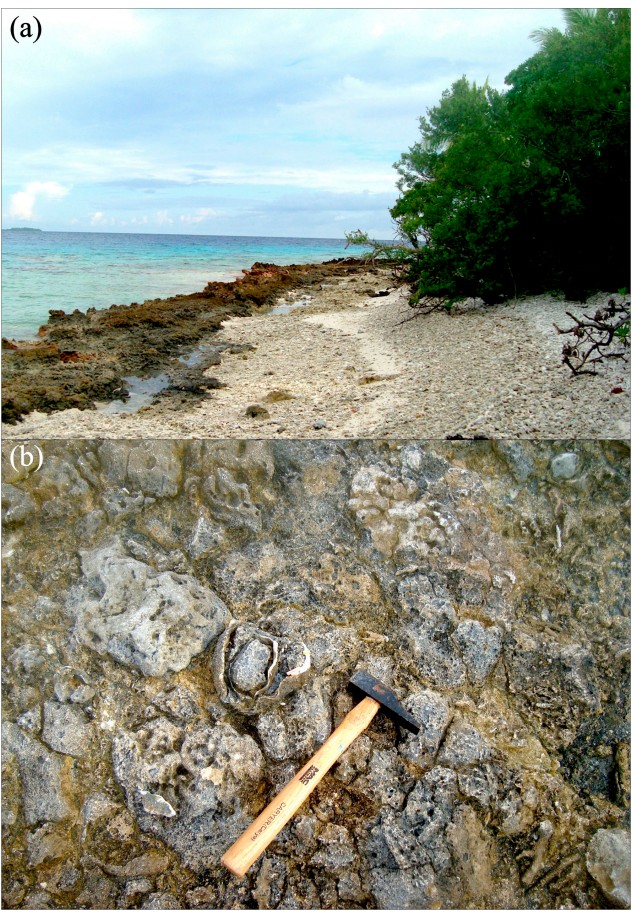

**Figure 6.** Lagoonal reef patch C1, Fakarava Atoll; see Figure 1f for location. (**a**) View of rubble ridge and conglomerate pavement, eastern side; (**b**) surface of the conglomerate pavement showing pebble-sized coral clasts and *Tridacna* shells.

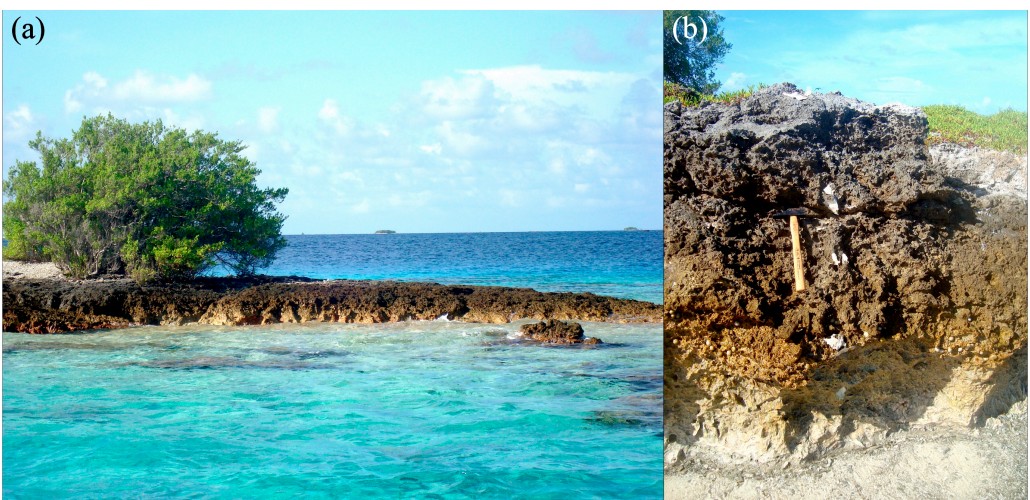

**Figure 7.** Lagoonal reef patch C2, Fakarava Atoll; see Figure 1g for location. (**a**) View of its northeastern end, showing the conglomerate pavement, up to 1 m thick, serving as an antecedent substrate for vegetated, sandy sheets; (**b**) section of the conglomerate platform—thickness: 1.30 m.

*3.2. Radiometric Dating Procedure and Age Significance*

The atoll islet depositional history was chronologically reconstructed using uranium series dating of 48 coral clasts, labelled FAK for samples collected in 2019 and F for samples collected in 2022. Sampling was conducted vertically along excavated sections and

superficially from selected adjacent conglomerate pavements. The procedure started with subsampling of the collected coral clasts using a micro saw. Thus, the most clean and pristine coral parts are extracted. These subsamples were rinsed with mQ water and ultra-sonicated several times. After adding a triple $^{229}$Th $^{233}$U $^{236}$U spike in a Teflon beaker, the subsamples were dissolved with diluted HCl. The U–Th separation and purification were performed after coprecipitation with Fe(OH)$_3$, on 0.6 mL columns filled with U-TEVA and prefilter resins, in nitric media—see Pons-Branchu et al. [88] for details. The U and Th isotopic compositions were analyzed at the Laboratoire des Sciences du Climat et de l'Environnement (LSCE, France), on a multi-collector inductively coupled plasma source mass spectrometer (MC-ICPMS) Thermo Scientific™ Neptune$^{plus}$ fitted with a desolvating introduction system (Aridus II), and a jet interface. For mass fractionation correction is used an exponential mass fractionation law—normalized to natural $^{238}$U/$^{235}$U isotopic ratio—and standard–sample bracketing. For more details on the analytical procedure— chemistry and MC-ICPMS analysis—see Pons-Branchu et al. [88]. After corrections for peak tailing, hydrate interference, and chemical blanks, $^{230}$Th/$^{234}$U ages were calculated (Table S1) from measured atomic ratios through iterative age estimation using the $^{230}$Th, $^{234}$U, and $^{238}$U decay constants [89,90]. Ages were corrected for detrital fraction using the value of the detrital fraction ratio $^{230}$Th/$^{232}$Th equal to $7 \pm 3.5$. Ages in the text and figures are expressed in calendar years before present relative to the year 2023. In order to make age results more easily comparable to previous research, particularly those based on radiocarbon measurements, ages are also given in conventional years BP, with the present referring to 1950 (Table S1).

Radiometric dating of displaced skeletal clasts for reconstructing atoll islet depositional histories were discussed previously [19,48,58–60,69]. Due to time uncertainty on clast deposition and stabilization, islet chronologies are regarded as relative, not absolute. Coral clasts may have been subjected to successive phases of reworking and deposition. This is clearly evidenced by a number of age inversions in the cross-sections (Figures 9 and 10). However, islet chronologies are assumed to be consistent, since most dated samples occupy a stratigraphic position within the cross-sections and conglomerates, which agrees with their age. Samples tend to be younger closer to the top and older closer to the bottom. In these sites, the youngest ages obtained are regarded to be closer to the time of definitive stabilization of the relevant islet section. Clast stabilization is believed to have taken place within 200–1000 yr long intervals, according to clast ages recorded at base of the sections.

## 4. Results

### 4.1. Islet Lithostratigraphy

Four major lithofacies were identified within the studied islets, primarily based on textural attributes.

### 4.1.1. Boulder–Pebble-Dominated Facies

Boulders—up to 50 mm in diameter—and pebbles—50–5 mm in diameter—are end-products mainly derived from pocilloporid, poritid, and merulinid colonies, together with large gastropod and pelecypod shells (Figure 8a). Throughout the studied sites, the amounts of pebble-sized sediments were markedly higher on the windward, eastern, than on the leeward, western rim sites. The mean diameter of the coral clasts ranges between 250–200 mm (fine boulder) and less than 5 mm (fine pebble) with highest values of 300 mm. Along the windward atoll side, the pebble sizes tend to decrease lagoonwards. Boulder–pebble-dominated facies grade lagoonwards into pebble, sand-supported, then in sand-dominated facies. The amounts of sand-sized sediments do not exceed approximately 10–15% of the total volume.

### 4.1.2. Pebble-Dominated Facies

The two subfacies can be differentiated by their clast-supported characteristics: pebble-supported and sand-supported subfacies. The first is present only within the windward,

ocean-facing section at different stratigraphic levels (Figure 8a), but are missing from the innermost windward and both leeward excavations, while the latter was reported from both atoll sides. Throughout the studied sites, the amounts of pebble-sized detritus were markedly higher on the windward than the leeward reef-rim sites. The sand and pebble-rich layers, about 5 to 30 cm thick, are distributed locally in the form of fining-upward, graded-bedding units.

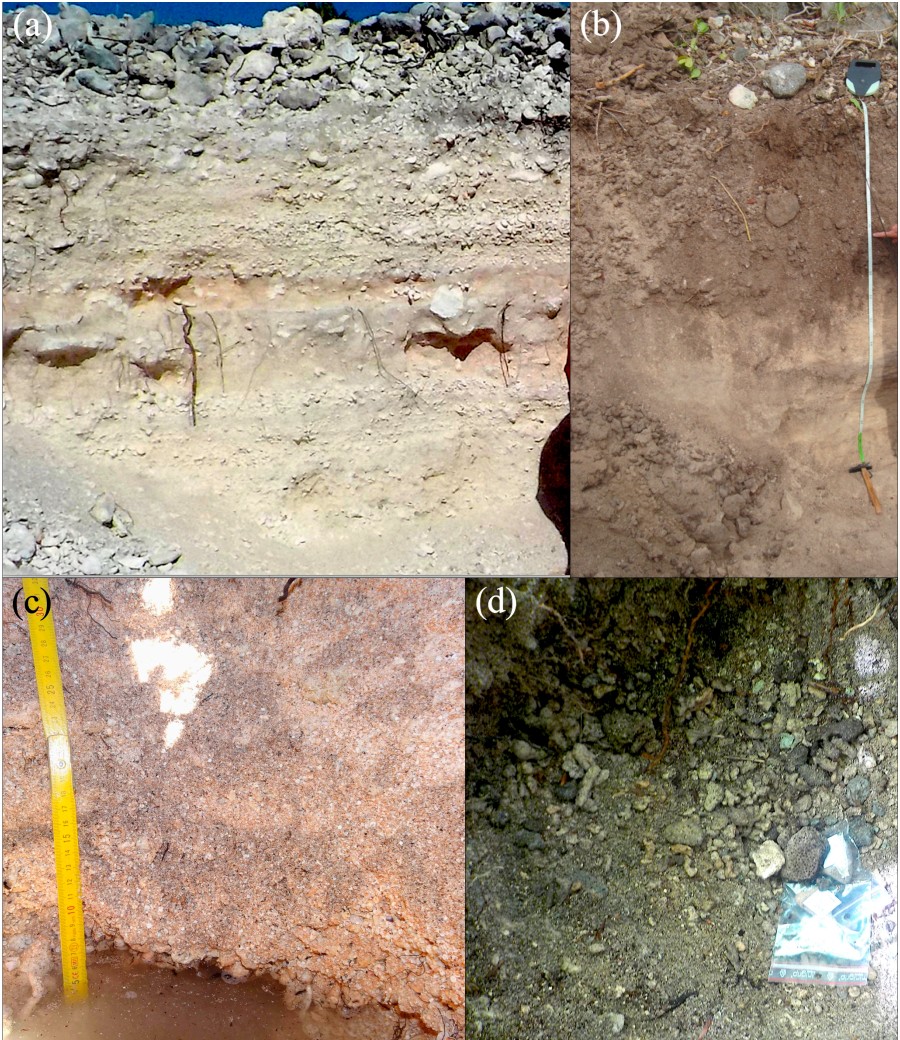

**Figure 8.** Views of lithostratigraphic sections through the reef-rim islets, Fakarava Atoll; see Figure 1c,d,e for location. (**a**) Windward, northeastern area: section A1, ocean-facing motu margin, about 2 m in total height, showing alternations of cobble- to sand-sized skeletal detritus; (**b**) windward, northeastern area: Section A2, center of the motu, about 1.70 m thick, showing sand-dominated deposits and an organic-rich layer at the uppermost section part; (**c**) leeward, southwestern area, Motu Topikite: Section B1, lowermost 0.40 m of the sand-dominated sequence; (**d**) leeward, southwestern area, Motu Otekofai, Section B2, bottom of excavation showing sand-supported, pebble detritus, rich in organic matter.

### 4.1.3. Sand-Dominated Facies

This was found in all excavated sites, from the ocean-facing to lagoon-facing margins at the windward, eastern reef-rim and within both the leeward, western rim islets. The layers are usually 0.20 to 0.50 m thick, with sands usually coarse-to-medium-graded (1 to 0.25 mm in mean size) and moderately to poorly sorted (Figure 8b–d). The proportions of sand and pebble-sized grains vary locally. In the sand-dominated depositional units, from both windward and leeward atoll sides, there is no apparent graded bedding. Together with

corals, coralline algal, and molluscan debris, the sandy fractions contain high amounts of benthic foraminifera, including amphisteginids—89–96% of the total foraminiferal fraction—and soritids.

### 4.1.4. Organic-Rich Facies

This facies relates to the occurrence of an organic-rich, brown-to-black-colored, root-bearing beds, usually located at the uppermost parts of the *motu* depositional units mainly at the expense of sand-supported and sand-dominated material [59,60]. At Fakarava, in the selected eastern atoll areas, sand-dominated deposits from the innermost excavations (Sections A2 and A3) are mixed with brown, organic-rich soils at different stratigraphic levels (Figure 8b). A similar feature is observed at Motu Otekofai (Site B2) throughout the whole sequence (Figure 8d). By contrast, organic-rich sandy material is missing from Motu Topikite probably because the local vegetation has a low life expectancy due to frequent islet reshaping.

On the windward, eastern atoll rim side, the cross-sections through the *motu* revealed that the top surface of the underlying conglomerate platform was found respectively at depths ranging between 2.50 and 0.65 m (Figure 9). Facing the ocean, Section A1, 2.50 m thick, exhibited alternations of pebble- and sand-dominated layers, varying from about 0.60 to 0.20 m in thickness. The sedimentary sequence from Section A2, 1.35 m thick, located in the middle part of the islet, is composed of sands rich in foraminifera, locally containing coral gravels and mixed with brown soils (Figure 8b). Section A3, 0.65 m deep, displayed sand-dominated detritus, mixed with organic-rich, brown soils. The grain-size decreases from the ocean-facing deposits lagoonwards. The pebble-dominated facies laterally grades into pebbly, sand-supported, then into sand-dominated facies.

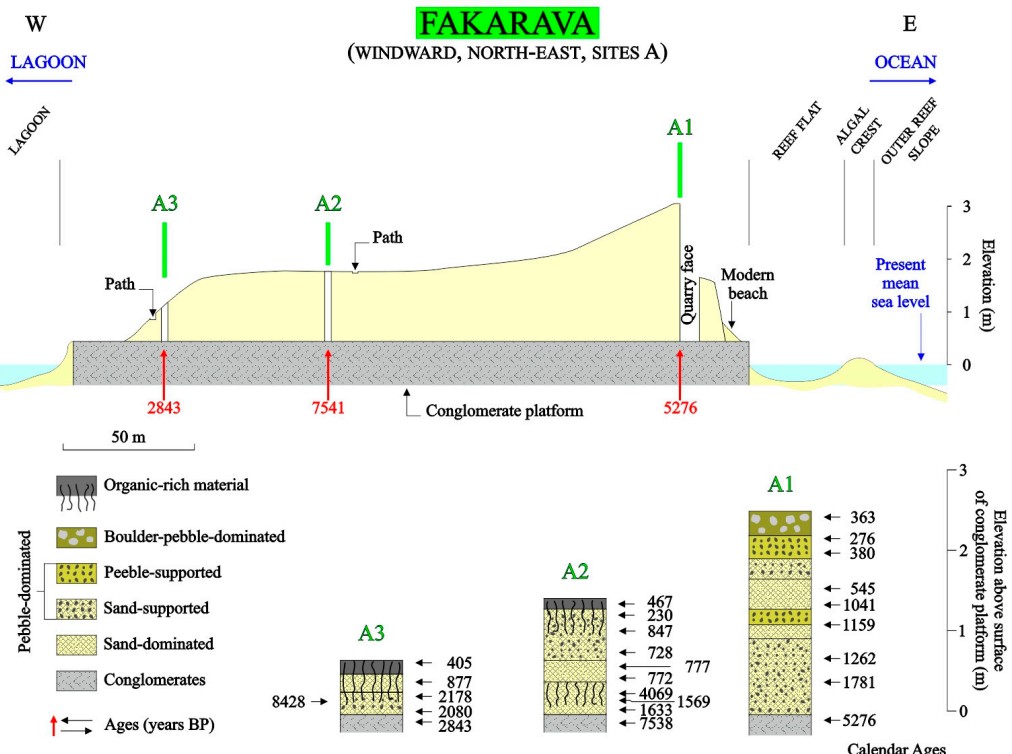

**Figure 9.** Transverse topographic profile across the windward, northwestern reef-rim *motu*, Fakarava Atoll, showing the location and lithostratigraphy of the three selected cross-sections A1, A2, and A3. The stratigraphic position of the U/Th dated coral clasts is indicated. See Figure 1b,c for section locations and Table S1, Supplementary Material, for dating results. Age are given in calendar years BP relative to 2023.

On the leeward, western atoll rim side, the excavated sections at Motu Topikite and Motu Otekofai are, respectively, 1 and 0.95 m thick (Figure 10). These are composed of sand-supported and sand-dominated material (Figure 8c).

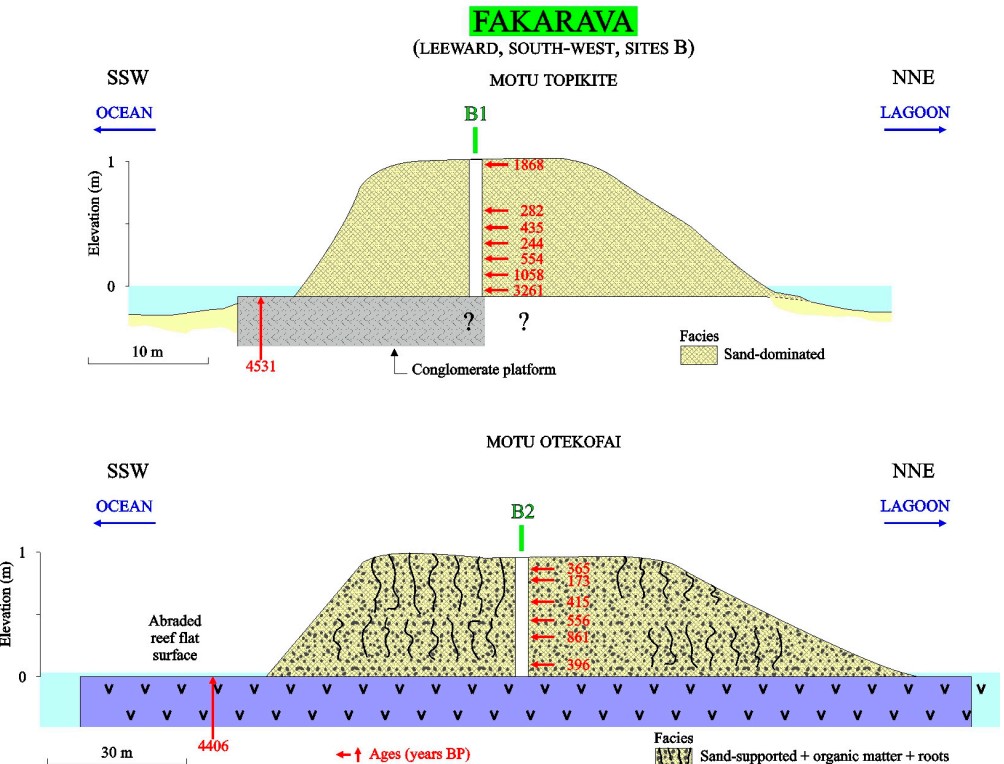

**Figure 10.** Transverse topographic profile across the leeward, southwestern reef-rim islets, Fakarava Atoll, showing the location and lithostratigraphy of the two selected cross-sections B1 (Motu Topikite) and B2 (Motu Otekofai). The stratigraphic position of the U/Th dated coral clasts is indicated. See Figure 1b,d and Figure 1b,e for section locations and Table S1, Supplementary Material for dating results. Ages are given in calendar years BP relative to 2023. The question mark (?) refer to uncertainty about the lateral extension of the conglomerate platform.

### 4.2. Islet Chronostratigraphy

At Fakarava, the coral clasts extracted from the conglomerate pavements, i.e., the foundations of reef-rim and lagoonal patch reef islets, returned ages of 7538 ± 67 BP (FAK 50), 5276 ± 21 BP (F 93), 4531 ± 16 BP (F 143), 2843 ± 12 BP (F90). At the lagoonal patch reef sites, conglomerates formed between about 5000 BP (FAK 58: 5020 ± 32 yr BP) and 700 yr BP (FAK 35: 702 ± 34 yr BP). The oldest age (FAK 50) was reported from the pavement underlying the middle part of the selected northeastern *motu* site, while the youngest one (FAK 35) comes from the lagoonal patch reef-supported islet (Reef Patch A, Motu Hakono). As a whole, beneath both the reef-rim and lagoonal *motu*, the average conglomerate ages vary between 5300 and 2900 yr BP along both the windward and leeward atoll sides. Accordingly, the dated conglomerate clasts clearly appear to be mostly older than those collected from overlying, unconsolidated detritus

In most cases, in the excavated unconsolidated depositional units, there is a common decreasing trend in the age from base upwards, consistent with stratigraphy (Figure 11). In the northeastern, windward *motu* sites, clast deposition appears to have started from about 2200 (F86A: 2171 ± 26 yr BP) or 2000 years ago (F86B: 2080 ± 10 yr BP) to 1600 years ago (FAK49: 1633 ± 15 yr BP; F97: 1781 ± 8 yr BP) and to have finished around 400–200 years ago (F113: 363 ± 5 yr BP; F74: 405 ± 3 yr BP; FAK39: 230 ± 8 yr BP). The average time lag between the base and top of each sequence, as a reference of the vertical accretion duration, can be estimated at between 1200 and 2000 years (Figure 11), considering yjr possible phases of islet reshaping. At the lowermost parts of the *motu* deposits, coral samples as

old as 4000 (FAK46: 4069 ± 17 yr BP), 5000 (F93: 5276 ± 21 yr BP), and 8000 years (F82: 8428 ± 57 yr BP) were found in the three excavations. These are interpreted as resulting from intense reworking during the early phases of *motu* accretion, thus allowing for clasts supplied from older stocks to be interbedded with more recent ones.

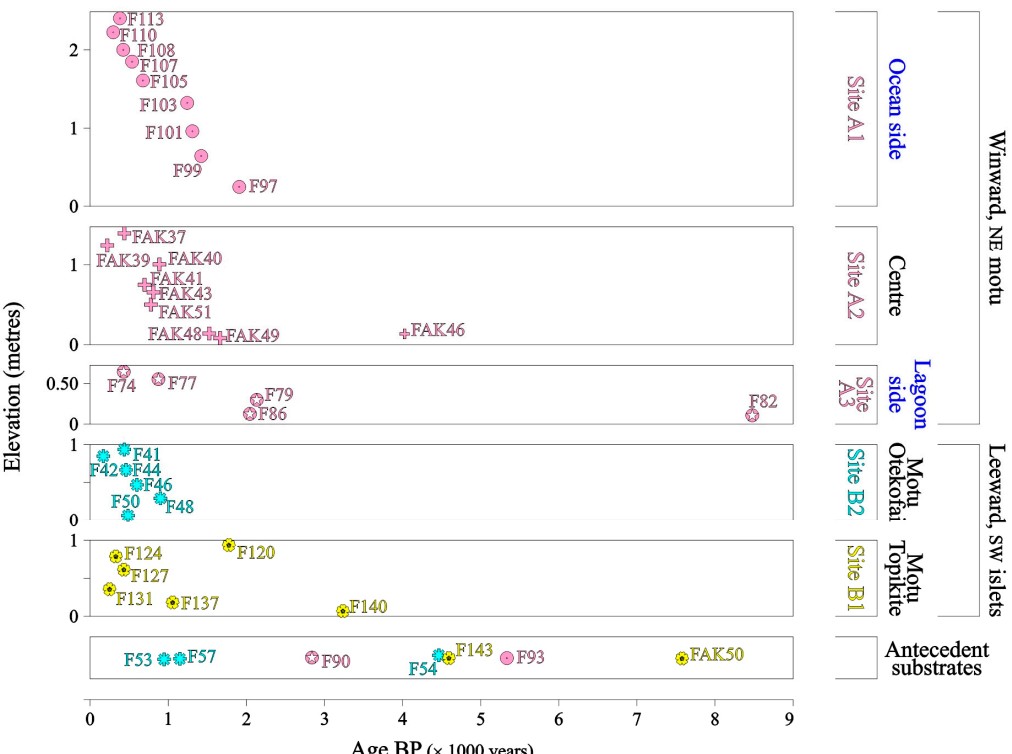

**Figure 11.** Plot of the U/Th dated coral samples versus their respective topographic and stratigraphic locations within the selected islet sites; see Figure 1a–g, Figures 11 and 12 for location. Age are given in calendar years BP relative to 2023. Note the decreasing age trend from base to top in each rim islet section.

A relatively different scenario emerged from the chronostratigraphical analysis of the southwestern islets (Motu Topikite and Motu Otekofai): the clast ages are around 1000–800 yr BP on average (F48: 861 ± 8 yr BP; F137: 1058 ± 12 yr BP), with the youngest ranging between about 400 and 170 yr BP (F44: 415 ± 23 yr BP; F131: 244 ± 7 yr BP; F42: 173 ± 3 yr BP). However, both the leeward sites are typified by a number of clast age inversions; some younger samples are located in the middle parts of the unit, not at the top. This indicates that these sandy cays have experienced relatively frequent sediment reworking and reshaping over the last 400 years. Moreover, the uppermost layer could contain detritus older than 1800 yr BP (F120: 1868 ± 19 yr BP). The time lag between the base and top of both cay sections ranges approximately between 250 and 700 years.

### 4.3. Islet Accretion Rates

The accretion rates of the islets can be assessed from the stratigraphic position of the radiometrically dated coral clasts collected within the excavated sections. The values obtained are just indicative, not absolute, since islet accretion is likely to have been periodically disturbed by sediment reworking and locally instantaneous re-deposition during marine extreme hazard events. The mean vertical accretion rates at the windward, northeastern *motu* sites decrease from the ocean-facing site lagoonwards, from about 2, 1, to 0.4 mm/yr. At the leeward, southwestern cay sites, the vertical accretion rates range around 1.4 mm/yr.

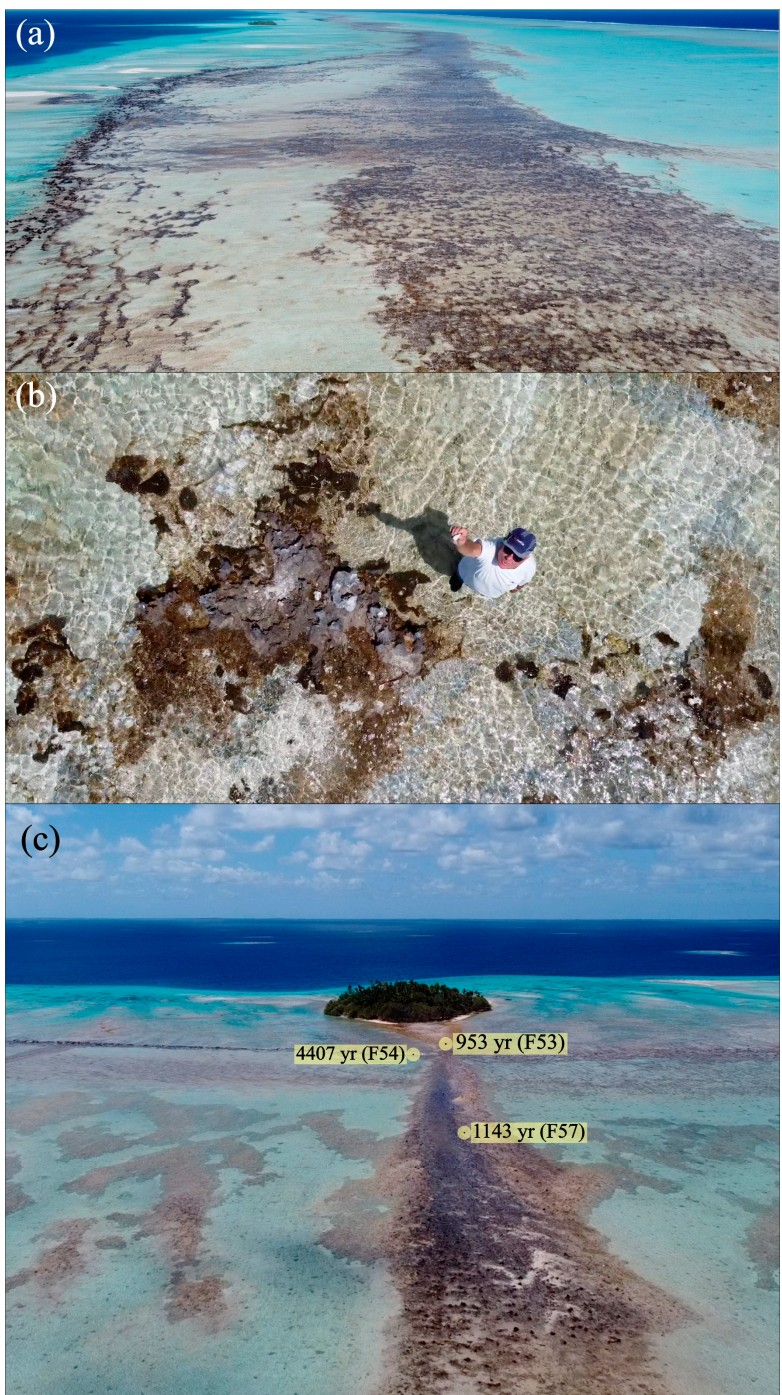

**Figure 12.** Drone aerial views of rubble and conglomerate sheet networks, leeward, southwestern, reef-rim, Fakarava Atoll. (**a**) Southwest of Motu Topikite, Site B1: about 50 m wide, band of partly eroded conglomerate pavement, parallel to the reef front line; (**b**) close-up of the eroded conglomerate pavement; (**c**) west of Motu Otekofai, Site B2: two alignments of conglomerate deposits, respectively oriented parallel and perpendicular to the reef front line and forming two distinct morphological patterns. The conglomerate pavement in planimetric continuity with the islet is about 20 m in maximum width.

## 5. Discussion

### 5.1. Timing and Mode of Deposition of Reef Detrital Material

At Fakarava, the oldest coral clasts incorporated into the rim deposits yielded ages of 7538 ± 67 yr BP (FAK50: conglomerates) and 8428 ± 57 yr BP (F82, Site A3, base of islet)

(Figure 11), indicating that the relevant coral colonies from which the clasts were extracted were living at times when the sea level was about 15 and 20 m respectively below its present position and, consequently, when the uppermost sections of the reef rim were being built. Coral detritus from the mid to early Holocene age were found to be trapped into the islet deposits in the nearby atolls of Rangiroa [58] and Takapoto [60]. Two alternative scenarios were invoked for explaining the trapping of such older coral fragments into the reef-rim sediments: step-by-step displacement with temporary deposition on forereef mid-depth terraces prior to being removed upwards; single-stroke transport upwards and transient accumulation within depocentres along the edges of the atoll rim tops prior to being removed inwards. In any case, this strongly suggests that rim deposits can experience frequent and severe reworking and clast enrichment from relatively distant sediment sources before final deposition and stabilization. The oldest age at the base of the ocean-facing site (F 93: 5276 ± 21 yr BP) may relate to a reworked clast from the initially deposited, underlying conglomerates.

### 5.1.1. Antecedent Substrates

As previously emphasized by [19,69,91,92], the presence of hard, pre-existing substrates, including coral reef flat or conglomerate surfaces, close to or at the sea level would be a prerequisite for islet initiation. At least, such indurated surfaces can promote retention of variously grain-sized detritus at specific nodal locations. At Fakarava, the main phases of conglomerate deposition appear to have occurred approximately over a duration of 7000 years, during the full mid to late Holocene sea level cycle. Over the last 2200 years, islets have been in the making, thus resulting in the interplay between the formation of conglomerates and emplacement of overlying unconsolidated sediments. This was demonstrated to be a common feature during the building of French Polynesia coral reef islands [69].

The most striking feature is the occurrence of continuous conglomerate bands along the leeward, southeast, innermost reef-rim zone at a constant distance of 900 to 100 m behind the reef front line. By contrast, conglomerate networks are weakly developed along the outermost rim zones, just forming localized, sub-circular to elongated patchy units, about 30 to 100 m wide and up to 250 m long (Figure 5). Field observations and the analysis of the drone images of the conglomerate networks of the reef area considered revealed that these pavements and adjacent reef flats, emplaced during the sea level high stand, between 4531 ± 16 yr BP (F143) and 4406 yr BP (F54) are being severely eroded (Figures 12a–c and 13). Similar conglomeratic alignments are observed along the leeward, western reef-rims of a number of nearby atolls, including Toau, Faaite, Tahanea, and Motutunga. Their location may reflect the limit of the transport of pebble-sized clasts by non-cyclonic, storm surges.

### 5.1.2. Islets

In the windward, northeast *motu* system, the clast age distributional patterns suggest that the three cross-sections have begun to emplace practically simultaneously, within a relatively short time lag of about 400 years (Figure 9). The sequence close to the oceanward shoreline (Site A1) commenced depositing around 1800 years ago (F97: 1781 ± 8 yr BP). Islet initiation at the innermost sites occurred at about 1600 yr BP (*motu* centre, Site A2, FAK49: 1633 ± 15 yr BP; FAK48: 1569 ± 10 yr BP) and 2200–2000 yr BP (lagoon margin, Site A3, F86A: 2080 ± 10 yr BP; F86B: 2171 ± 26 yr BP). The age comparison between the uppermost layers of the lagoon-facing section and of the outermost ones gives support to the idea that the deposits of the lagoon margin have remained stable over the last 400 years (Site A3, Sample F74: 405 ± 3 yr BP), while the ocean-facing margin as well as the central part of the *motu* bench have suffered intense reshaping over the two last centuries (Site A1, F110: 276 ± 2 yr BP; Site A2, FAK39: 230 ± 8 yr BP). Finally, the timeline of the distributional pattern reflects an accretion that would have started approximately 2000 yr

ago and remained active until about 200 yr BP (Figure 14), probably prior to the relative *motu* stabilization by sustainable vegetation establishment.

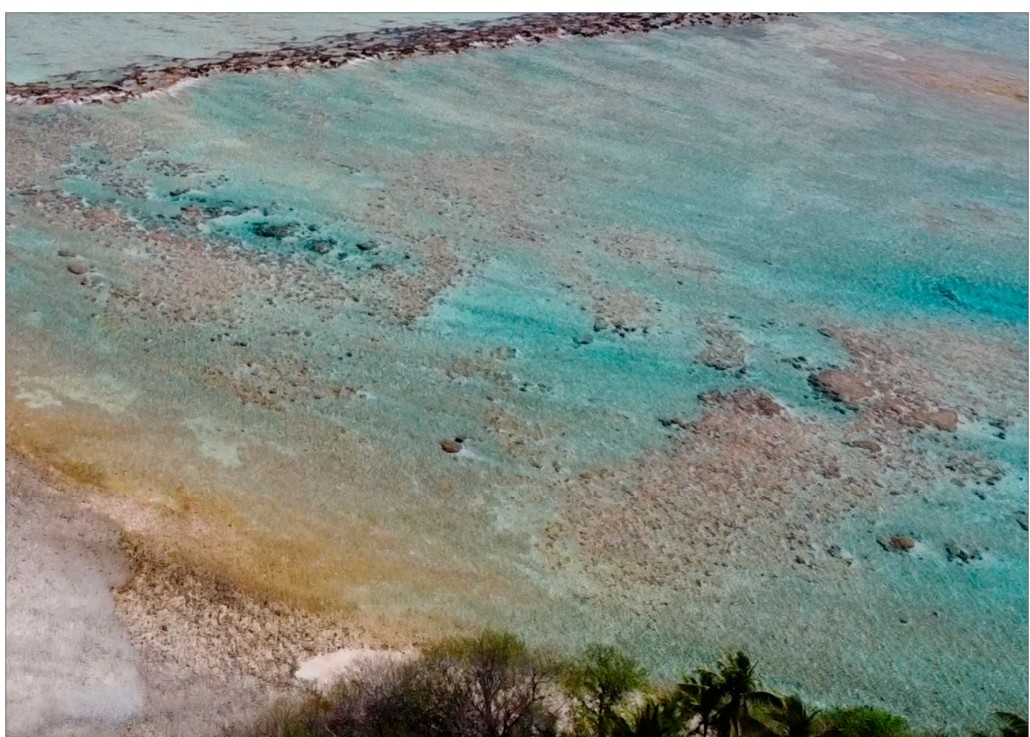

**Figure 13.** Drone aerial view of the innermost reef flat zone, northwest of Motu Topikite, showing the abraded reef surface behind a conglomerate alignment.

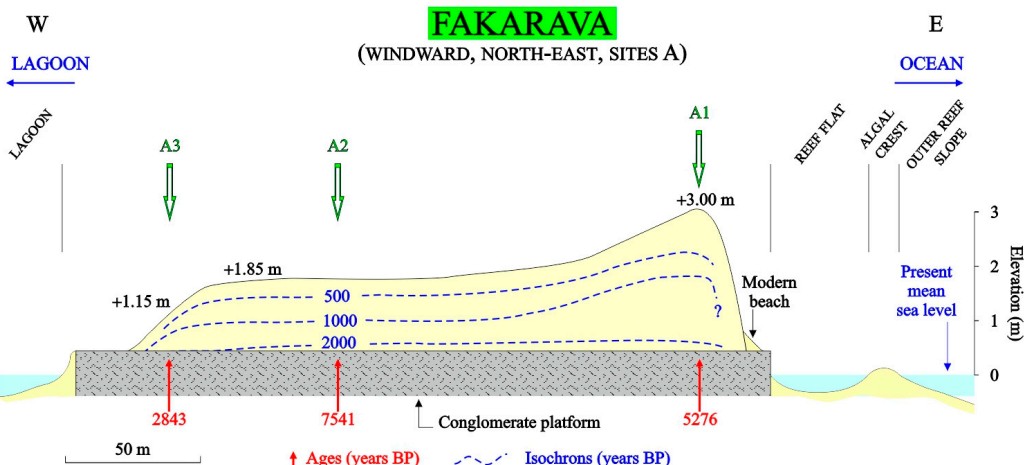

**Figure 14.** Cross-section through the windward, northwestern reef-rim *motu*, Fakarava Atoll, showing the successive stages of *motu* building. The dashed lines refer to timelines, given in calendar years BP (relative to 2023), and are delineated from time intervals between the dated coral samples. The location of the three cross-sections—Sites A1, A2, and A3—and that of coral clasts extracted from the basal conglomerate pavement are given. The question mark (?) refers to uncertainty about the lateral extension of isochrons.

Along the leeward, southwestern atoll side, Motu Topikite and Motu Otekofai have settled respectively on a conglomerate platform, 4531 ± 16 yr BP old (F143), and on a partly eroded reef flat, 4406 ± 34 yr BP old (F54). At the base of Motu Topikite, the sequence exhibits older material, dated 3261 ± 43 yr BP (F140). By contrast, at the base of Motu Otekofai, an age of 396 ± 33 yr BP (F50) was found. At Motu Topikite, the older

basal clast may have resulted from an earlier deposition and stabilization, related to the original depocenter from which the studied cay has expanded. A Motu Otekofai, the younger basal clast can be related to a periodically deep reshaping of the islet, destruction and reconstruction, thus allowing for clasts derived from the youngest sediment stocks or conglomerates to be removed and re-deposited at the base. Although, at the base of both the excavated sections, there are older coral clasts, the age distributional patterns in these sections revealed that the main vertical aggrading phases started about 1000 years ago (F137: $1058 \pm 12$ yr BP) at Motu Topikite and 800 years ago at Motu Otekofai (F48: $861 \pm 8$ yr BP).

When compared to that of other studied reef-rim islets from the Tuamotu atolls [12,58–60], the mode and timing of the islet accretion at Fakarava refers to a different dynamic model. The analysis of the internal architecture from the northeast *motu* indicates that apparently no distinct nodal point, i.e., depocenter, from which *motu* accretion would have initiated and spread out oceanwards and lagoonwards, was identified. Its development seems to have arisen relatively coevally throughout the whole depositional system, even if the ocean-facing side appears to have accreted more rapidly than the inner *motu* parts. The islet accretionary model presented herein refers to the *vertically episodic accreting* model defined by [19], in which accretion starts relatively coevally at several points from the ocean-facing to lagoon-facing reef-rim top surface and extends upwards episodically through time.

In addition, islet initiation appears to have begun much later. By contrast, on Takapoto Atoll, in the northwest Tuamotu region, islets from both windward and leeward rim sides started to accrete around 2500 years ago from the initial depocenters located in the mid rim areas, then continued to grow laterally towards both the ocean- and lagoon-facing margins [59,60]. At South-Marutea Atoll, at the southeastern Tuamotu end, the reef-rim islets at both windward and leeward sittings initiated much earlier, about 5000 years ago, and originated from depocenters located along the ocean-facing rim side, then prograded towards the lagoon side [12]. At Rangiroa, the preliminary results revealed that, in the northeastern atoll rim zone, *motu* building is likely to have started as soon as 6000 years ago and achieved about 3000 years ago [58]. The timing of islet building in the considered windward and leeward Fakarava sites appears to be very similar to that reported from Tepuka Island, on Funafuti Atoll—Marshall Islands, Central Pacific [36]. The island of Tepuka began to build at about 1100 yr BP and spread out over the following 500 years. Islet accretion on both Fakarava and Tepuka, at least in the considered areas, began much later than in a number of other low-lying coral islands in the Indo-Pacific Province. In the Maldives (western Indian Ocean), islet deposition initiated around 5000 years ago [46] or between 2000 and 500 yr BP [49] according to the considered areas. Similarly, in the Marshall Islands, on Jabat Island, deposition commenced at around 4800 yr BP and was active for about 800 years [48].

There was a marked retardation in islet initiation as compared with the age of reef flat emplacement. The dating of a coral sample extracted from the leeward, southwestern, inner reef flat provided an age of $4406 \pm 19$ yr BP (F54). This age agrees with those from other Tuamotu reef flat locations. At Moruroa Atoll (southern Tuamotu), reef flats have grown near sea level since 4000–3000 yr BP [93]. At Fangataufa, a nearby atoll, the dating of the cored coral reef flat returned calibrated radiocarbon ages of $4315 \pm 340$ yr BP at a maximum depth of 4.1 m below pmsl (Montaggioni, unpublished), strongly suggesting that the reef flats had caught up with sea level not before 4300 yr BP. This indicates that most French Polynesian reef flat top surfaces are likely to have reached their present position while the sea level was about 0.80 m above the present mean sea level [71]. A minimal duration of about 1700 years, from 4300 to about 2000 yr BP, may have separated the final phase of the reef flat aggradation from the beginning of the islet building, at least at the excavation sites. The presence of coral clasts older than 3000 yr BP at the lowermost parts of the Motu Topikite and Motu Otekofai sections could be indicative of an earlier clast deposition and incipient island accretion prior to the intensive reworking and resettlement of more sustainable sedimentary bodies over the last 1000 years.

The main depositional episodes of the islets at both the selected windward and leeward rim settings occurred over the last 2000 years (Figures 9 and 15) as the sea level dropped from about +0.50 above pmsl to its present position [71]. At Takapoto, *motu* deposition took place from 2500 yr BP during the drop in the sea level [59,60]. In this scope, such islet accretionary scenarios agree well with [94], suggesting that coral reef islet formation was promoted by a gradual sea level fall. By contrast, in other Tuamotu atoll locations, including Rangiroa and South-Marutea, *motu* development initiated during the overall sea level rise 6000 to 4000 years ago and continued to be active during the following high stand (4100–3500 yr BP) and afterwards, until the last few centuries [12,58]. These data confirm that islet formation may have occurred at any time during the mid-late Holocene sea level course, during lower or higher still-stands, and during the rising or falling sea levels as well. These scenarios differ from some others, especially those established from the tropical regions devoid of cyclones. The differing models assume that the initiation of low-lying coral reef islets was typically driven by marine transgressive and high-stand events, thus reactivating the high-energy window [36,46,52,95]. As formed within a variety of sea level contexts, islets will respond differentially to the predicted sea level rise. Those, such as the ones described in the present paper and emplaced during marine regressive episodes, may be disturbed by the current rising sea level [1,96], especially if the sea level rise is expected to occur at rates up to 8 mm/yr in the Tuamotu [86], storminess is expected to increase in intensity [14,97–100] and vertical reef-growth rates are expected to not exceed 4 to 5 mm/yr on average [101] in the next few decades. Accordingly, at least, in cyclone-prone areas, the sea level change may not be a significant control on atoll islet settlement and growth.

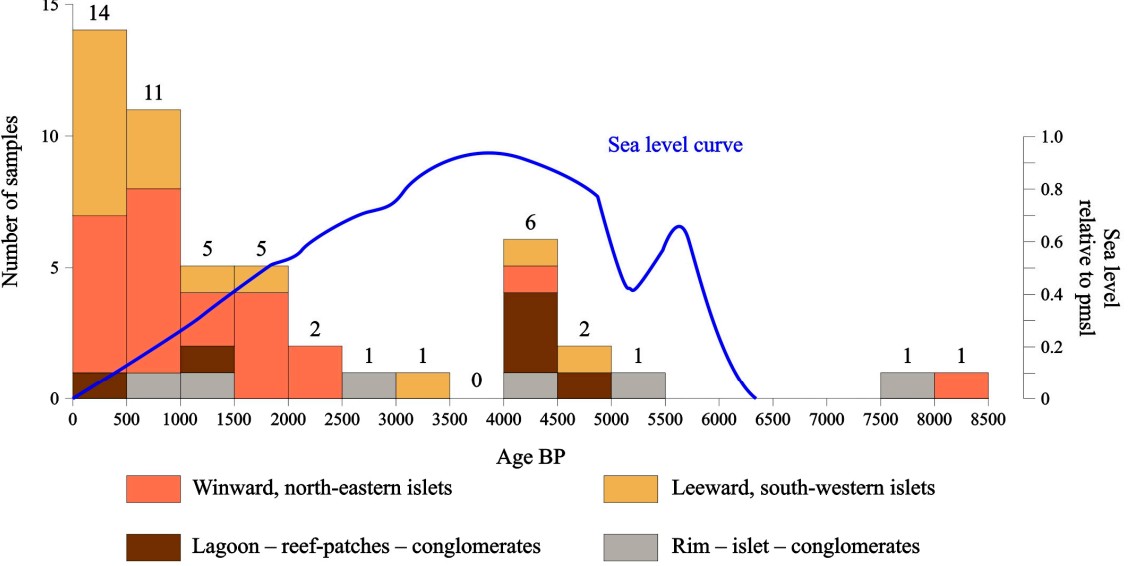

**Figure 15.** Age distributional pattern of U/Th dated coral clasts extracted from all the selected reef-rim and lagoonal patch reef sites at Fakarava, relative to sea level change over the last 6000 years in the Tuamotu region. The pattern is expressed by the number of counted clasts per 500 yr slice. Note that islet accretion on both atoll reef-rim sides occurred predominantly during the last 2000 years, as sea level was dropping to its present position.

### 5.2. Rates of Islet Accretion

The rate of islet accretion is known to mainly depend on the intensity and frequency of storm surge events, acting as both constructional and erosional agents, which have hit atolls in the past [19,63] on the turnover of adjacent sediment stocks [12,58,60]. The main source of coral clasts for supplying reef-rims are upper forereef coral communities [78], which are able to regenerate after 20 years in high-energy settings, significantly less than the apparent return period of islet formation events of about 50 years or more [14].

Extreme hydrodynamic events can be poorly, partly, or not preserved in islet stratigraphies, especially when sedimentary bodies are predominantly composed of fine-sized (sands and fine pebbles), mostly mobile detritus, as observed in the selected leeward sites at Fakarava. The calculated vertical accretion rates at both the windward and leeward excavation range from 4 to 20 cm per century. Such values are relatively close to those estimated from other Tuamotu atoll rim islets: Takapoto: 10–50 cm per century [60]; South-Marutea: 5–20 cm per century [12]. In the Maldives, in a region devoid of cyclones, vertical islet accretion rates are relatively similar to those obtained from the cyclone-prone Tuamotu region: 15–30 cm per century from atoll reef-rim units [52] and 27–30 cm per century from adjacent lagoonal islets [49]. These results suggest that at Fakarava, and probably in all the northwestern Tuamotu islands, in addition to cyclogenesis, winter storminess and moderate climate events may play a significant in the control of islet morphodynamics [15].

### 5.3. Role and Frequency of Marine Hazards Events

Within the windward, northeast *motu*, except at the ocean-facing site, the excavated sedimentary sections showed that sand- and pebble-sized material is predominant over cobbles. The leeward, southwest islets were shown to be composed mostly of foraminiferal sands. This seems to confirm that the main phases of islet formation have operated, if not under fair weather conditions, at least under moderate climate hazards, such as winter storms, and low and moderate tropical depressions, rather than under typical cyclonic events. Indeed, cyclones, as constructional agents, preferentially promote the motion of cobbles, large blocks, and mega-blocks [63,102], even if finer-grained detritus are also displaced from adjacent sedimentary sources.

Accordingly, the dating of pebbly clasts can contribute to providing valuable information on the frequency of storm swells that have impacted islands [103]. Based on the 33 U/Th dates covering the last 2000 years, i.e., the common era, and considering possible date overlapping, a maximum number of 3–5 marine hazard events per century was identified, mainly concentrated in the last millennium, probably due to biases in the data collection from the older time range, or promoted by intensified cyclogenesis during the Little Ice Age [76] (Figure 16). This is consistent with previously published historical records [77,104]—Canavesio, pers. comm.—and event reconstructions from clast dating [12,59,60], revealing that the number of marine high-energy hazard events would have rarely exceeded three to four per century over the last millennia. It is noteworthy that tsunami-generated wave surges in the Tuamotu are of low frequency, about six since the 16th century [63], and of low amplitude, not exceeding 2 m high [82]. Accordingly, the role of tsunamis in *motu* shaping is regarded as negligible in the considered region.

### 5.4. Comparaison with Other Coral Reef Islets

The reconstruction of low-lying, atoll-islet development in other Tuamotu atolls, including Takapoto [59,60], Rangiroa [58], and South-Marutea [12], indicates that the depositional mode and timing differ from atoll to atoll. *Motu* formation on a given atoll refers to a specific model according to the atoll island classification [19]. At Takapoto, both studied windward and leeward *motu* have accreted according to the *central core* model in which islet accretion starts from depocenters (i.e., central accreting loci) and extends both oceanwards and lagoonwards through time. Islet initiation started at around 2500 years ago. By contrast, in northeast Rangiroa, the preliminary analysis revealed that the islet would have been emplaced as soon as 6000 yr BP. At South-Marutea, in both the windward and leeward *motu* settings, the accretionary model refers to the *regular lagoonward accretion* model, in which accretion started from depocenters located close to the ocean-facing side and then has incrementally extended preferentially lagoonwards over the last 5000 years. By comparison, the internal structure of the windward, northeastern *motu* at Fakarava has

resulted from a *regular, vertically accreting* mode, according to [19]. This *motu* was initiated just before 2000 yr BP and, thus, is the youngest identified at the regional scale.

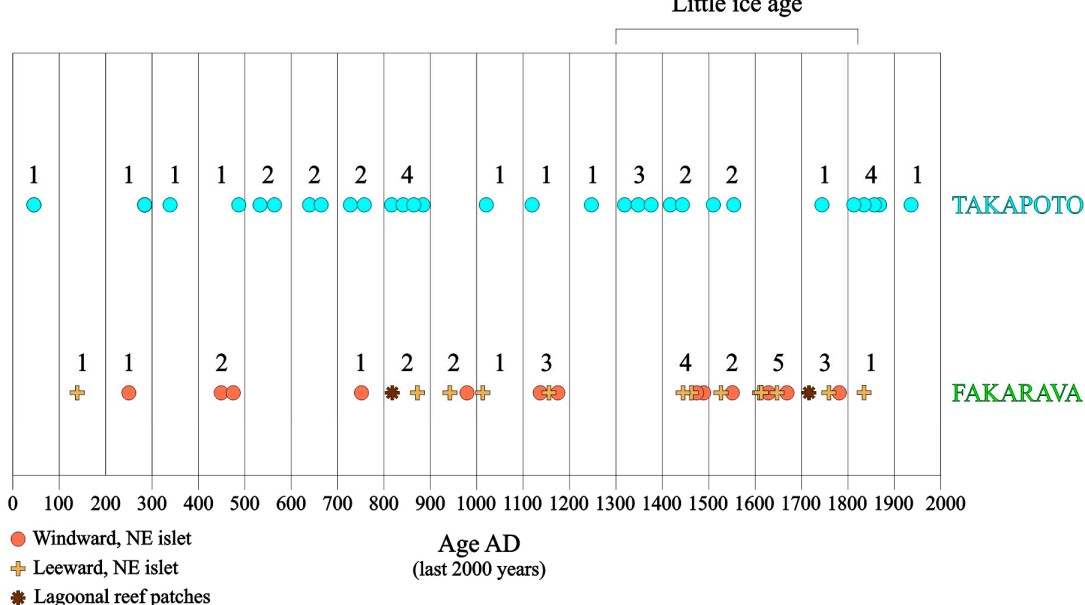

**Figure 16.** Age distribution of marine hazard events identified at Fakarava Atoll throughout the last 2000 years, based from U/Th dating of collected coral clasts in all the selected reef-rim and lagoonal reef-patch sites. The ages of occurrence of these events are compared with those obtained at Takapoto Atoll, northwestern Tuamotu, based on a similar methodological approach [60]. In both atolls, the frequency of hazard events does not exceed 3–5 per century. The blue circles refer to marine hazard events recorded from dated coral clasts in the islet deposits at Takapoto atoll.

A number of atoll islet formation models have been previously established from reef sites in the Indo-Pacific domain, assuming that the long-term coral reef islet stability is mainly controlled by sea level changes [19,36,46–49,52,55,91,92,105–108]. The elevation of the antecedent substrates—e.g., reef flats, conglomerate platforms, lagoonal hard to soft surfaces—close to or at sea level at the time of *motu* initiation appears to be critical to *motu* initiation and expansion. In French Polynesia, the delay in reef rim-islet initiation is directly driven by the time of the availability of conglomerate pavements serving as antecedent foundations. Islet initiation usually occurred while conglomerate genesis was still fully active for several centuries [69]. In addition, the previous studies on atoll islet accretion in the Tuamotu have revealed that the position of the sea level relative to reef surfaces during its course has apparently no direct control on the mode and duration of deposition and on the amount of deposited material. Islet development could have occurred at different times, during the rise or the fall in the sea level, resulting in differently sized *motu* [12,60].

Differences in the accretional mode and initiation time of islets could be explained by their respective location relative to the trajectories of extreme hazard events and rim exposure to storm swells, by their frequency and intensity, and by the availability of adjacent sediment reef stocks. In the considered region, including Fakarava and the nearby atolls (Faaite, Tahanea, Motutonga), *motu* are specifically restricted along the northern and eastern rim areas, open to winter east–northeast storms about 70% of the year. By contrast, the frequency of cyclones and associated events over the past 50 years was as follows: a cyclone, a strong tropical storm, and three moderate ones [75]. Similarly, although the western and southern atoll shores at Fakarava are widely open to distant-source storm surges, there is no continuous, elongated islets, but only a series of small-sized, usually isolated sandy islets, covered or not with vegetation, together with networks of unconsolidated to lithified shingle sheets. This strongly suggests that the generated wave surges may have crossed

over reef flat zones, causing erosion rather than detritus deposition and, as observed on Makemo Atoll in 1996 and 2011, a rise in the la goon water level and submersion [14]. Such events in the past may have resulted in the formation of conglomerate pavements atop of lagoonal reef patches. As a summary, the *motu* from the windward, eastern atoll sides would have been generated by winter storms and occasional cyclonic and depressional events, while the leeward, western and lagoonal geomorphic features would be rather shaped by periodical distant-source swells.

## 6. Conclusions

On Fakarava Atoll, along the windward, northeast reef-rim, the main episodes of conglomerate emplacement occurred between 5500 and 2500 yr BP during a time interval including both rising and falling sea level movements. The overlying unconsolidated detritus commenced accreting by about 2000 yr BP while the sea level was gradually dropping from +0.50 m above pmsl to its present position. This confirms that the control of the sediment nourishment of reef flat surfaces did not depend on the sea level position. Islet building appears to have been governed by changes in the intensity and frequency of marine hazard events rather than sea level fluctuations over time. No specific depocenter from which the windward, northeast *motu* could have expanded was identified. This may suggest that this *motu* as a whole has accreted more or less synchronously across the whole rim surfaces. Along the southwest rim side, the selected islets, still experiencing positional change, have been emplaced mainly over the last millennium, but resting on antecedent hard substrates as old as approximately 4500 yr BP.

Islets from other Tuamotu atolls have accreted under different modes and past sea level positions. Such intra-regional contrasts probably will result in a variety of different behavioral responses to climate change from atoll to atoll. Accordingly, in the current context of the rising sea level and increasing storminess, defining the physical trajectories of low-lying islets in the near future remains speculative. At Fakarava, since these bodies have formed under falling sea level conditions, any rise in the sea level and opening of higher-energy windows could result in rim overflooding and islet instability.

**Supplementary Materials:** The following supporting information can be downloaded at https://www.mdpi.com/article/10.3390/geosciences13120389/s1, Table S1: Uranium–thorium data of coral samples from Fakarava.

**Author Contributions:** Conceptualization: L.F.M. and B.S.; methodology (field work): B.S., G.P., É.B. and M.T.; (laboratory analysis): L.F.M.; (U/Th dating): E.P.-B. and A.D.; validation: L.F.M. and B.S.; formal analysis: L.F.M. and B.S.; writing—original draft preparation: L.F.M., B.M.-G. and B.S.; writing—review and editing: L.F.M., B.M.-G., B.S. and G.P.; graphics: B.M.-G.; supervision: B.S., L.F.M. and B.M.-G.; project administration and funding acquisition: B.S. All authors have read and agreed to the published version of the manuscript.

**Funding:** This research was partly supported by the research project no6538/MED/REC of 16 September 2019—Délégation à la recherche, Tahiti, leaders: Jean Yves Meyer and Serge Planes. Additional funding came from the research project no9553/MCE of 12 August 2021—Direction de l'environnement, Tahiti, MT—supporting major field research (BS, EB, GP) in August–September 2022.

**Data Availability Statement:** The data can be obtained from the leading writer (montaggioni@cerege.fr).

**Acknowledgments:** We warmly thank the following colleagues for their support: Jean-Yves Meyer, Serges Planes, the research project nº9553/MCE of 12 August 2021—Direction de l'environnement, Tahiti, MT—supported major field research (BS, EB, GP) in August–September 2022 and Annie Aubanel for field collaboration in March 2019—excavation through the windward *motu*, Site A2. This agreement allowed us to be able to prospect islets located in the protected Biosphere Reserve.

**Conflicts of Interest:** The authors declare no conflict of interest.

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
