# Peer review of "Holocene Depositional History of Low-Lying Reef-Rim Carbonate Islets of Fakarava Atoll, Northwest Tuamotu, Central South Pacific"

_geosciences, doi:10.3390/geosciences13120389_

Round 1

Reviewer 1 Report

Comments and Suggestions for Authors

Dear Authors,

I read your manuscript and I think that is well written and well documented. This fill some gaps in our knowledge of tropical islet developments.

I have very few comments in the attached file and I just suggest to add some figures to better visualise the place of some reported localities.

Best regards,

Author Response

Reviewer # 1

I have very few comments in the attached file and I just suggest to add some figures to better visualise the place of some reported localities.

All the suggested corrections are taken into account. All the studied excavated sites are clearly localized by the provided sites. In fact, during the manuscript uploading, the figures showing the excavated cross-sections (Figures 8a to 8d) have disappeared. This is now corrected.

Reviewer 2 Report

Comments and Suggestions for Authors

Montaggioni et al Fakarava Atoll.

This is an interesting and generally well written paper. The research is based on sampling sections through coral islands from both windward and leeward sites; dating the deposits and comparing these with the recorded sea-level history of the region as well as the history of extreme weather events. The main conclusions reached are that trajectory and exposure to extreme weather events is a more important control on island accretion than is the sea-level history. This is contrary to previous published work.

For these reasons I recommend publication after minor amendments as suggested in the attached pdfs. One for the text and the other for the figures.

Abstract and Introduction- Clear and well written. Minor improvements to Englishislands suggested- see edits on reurned marked up pdfs.

2. Setting

What is the datum? Maximum elevation above what?? In Intro and l.102

l. 116-118 “Conglomerate, needs some  clarification. I assume coral clasts so its coral conglomerate (as per l. 140) or coral rudstone??

l. 142 “high value” in what sense??

Fig 2. Make it clear that this graph is recording the direction from which winds are blowing from. And not to) to be consistent with text. This is a French versus English thing!

Fig 4. Trajectories need arrows?

l. 163. Meaning of “high” and Fig 3 needs to state that this records the  direction from which swells are arriving from. And not to) to be consistent with text.

l.185 Who or what is Canavisio

3. Materials and Methods.

l. 230-231 and l.276. Make it clear if these are transported clasts or whether any in situ coral was sampled. This is important.

l. 235 Add Rake to fig 1.b.

l.245 “sequence analysi”. Explain, what does this mean??

l.298. Table 1 not supplied in pdf.

4. Results

l. 317 No Fig 8 supplied in pdf.

Fig 10. Purple unit needs explanation

5. Discussion

l. 453 Use of term “sequence”. Is this being used in general term as in “depositional unit “ or “succession” or in sequence stratigraphic sense as a “depositional sequence”. I think it’s the former on which case use another word so as to avoid confusion. Also chaeck earlier usage in paper.

@font-face {font-family:"Cambria Math"; panose-1:2 4 5 3 5 4 6 3 2 4; mso-font-charset:0; mso-generic-font-family:roman; mso-font-pitch:variable; mso-font-signature:3 0 0 0 1 0;}@font-face {font-family:Calibri; panose-1:2 15 5 2 2 2 4 3 2 4; mso-font-charset:0; mso-generic-font-family:swiss; mso-font-pitch:variable; mso-font-signature:-536859905 -1073732485 9 0 511 0;}p.MsoNormal, li.MsoNormal, div.MsoNormal {mso-style-unhide:no; mso-style-qformat:yes; mso-style-parent:""; margin:0cm; mso-pagination:widow-orphan; font-size:12.0pt; font-family:"Calibri",sans-serif; mso-ascii-font-family:Calibri; mso-ascii-theme-font:minor-latin; mso-fareast-font-family:Calibri; mso-fareast-theme-font:minor-latin; mso-hansi-font-family:Calibri; mso-hansi-theme-font:minor-latin; mso-bidi-font-family:"Times New Roman"; mso-bidi-theme-font:minor-bidi; mso-fareast-language:EN-US;}.MsoChpDefault {mso-style-type:export-only; mso-default-props:yes; font-family:"Calibri",sans-serif; mso-ascii-font-family:Calibri; mso-ascii-theme-font:minor-latin; mso-fareast-font-family:Calibri; mso-fareast-theme-font:minor-latin; mso-hansi-font-family:Calibri; mso-hansi-theme-font:minor-latin; mso-bidi-font-family:"Times New Roman"; mso-bidi-theme-font:minor-bidi; mso-fareast-language:EN-US;}div.WordSection1 {page:WordSection1;}

Comments on the Quality of English Language

Author Response

Reviewer # 2

Abstract and Introduction- Clear and well written. Minor improvements to English islands suggested- see edits on returned marked up pdfs.

Corrections are taken into account.

2. Setting

What is the datum? Maximum elevation above what?? In Intro and l.102

The term datum seems to be confusing. We have left out it. In fact, it was used as a synonym of level.

  1. 116-118 “Conglomerate, needs some clarification. I assume coral clasts so its coral conglomerate (as per l. 140) or coral rudstone??

We have distinguished between unconsolidated and lithified coral clasts. Only the second types are regarded as conglomerates. By definition, a conglomerate is composed of CEMENTED grains and, consequently, unconsolidated material cannot be regarded as conglomerate.

  1. 142 “high value” in what sense??

Because endemic, high-value in terms of biodiversity.

Fig 2. Make it clear that this graph is recording the direction from which winds are blowing from. And not to) to be consistent with text. This is a French versus English thing!

Any wind rose (as swell rose) directs toward the zone where winds and swells are coming from. This is not a French but an international agreement.

Fig 4. Trajectories need arrows?

Done.

  1. 163. Meaning of “high” and Fig 3 needs to state that this records the direction from which swells are arriving from. And not to) to be consistent with text.

It means high latitude. Corrected.

l.185 Who or what is Canavisio

Canavesio is a geographer, working on the climate in French Polynesia.

  1. Materials and Methods.
  2. 230-231 and l.276. Make it clear if these are transported clasts or whether any in situ coral was sampled. This is important.

As clearly indicated, all the dated coral collected are clasts, not corals in growth position.

  1. 235 Add Rake to fig 1.b.

This location name relates to the area designed as A in Figure 1b.

l.245 sequence analysis. Explain, what does this mean??

This means lithological analysis of the depositional unit. Corrected.

l.298. Table 1 not supplied in pdf.

Table 1 (dating results) is added. Disappeared during the initial uploading of the manuscript.

  1. Results
  2. 317 No Fig 8 supplied in pdf.

Figure 8 has disappeared during the initial uploading. Corrected.

Fig 10. Purple unit needs explanation

  1. Discussion
  2. 453 Use of term “sequence”. Is this being used in general term as in “depositional unit or succession or in sequence stratigraphic sense as a “depositional sequence”. I think it’s the former on which case use another word so as to avoid confusion. Also check earlier usage in paper.

To avoid any confusion, the term sequence is replaced by depositional unit or section.
